# Recycling Non-Metallic Powder of Waste Printed Circuit Boards to Improve the Performance of Asphalt Material

**DOI:** 10.3390/ma15124172

**Published:** 2022-06-12

**Authors:** Sheng Li, Yu Sun, Shuo Fang, You Huang, Huanan Yu, Ji Ye

**Affiliations:** 1Key Laboratory of Special Environment Road Engineering of Hunan Province, Changsha University of Science & Technology, Changsha 410114, China; lishengttt@163.com; 2School of Traffic and Transportation Engineering, Changsha University of Science & Technology, Changsha 410114, China; s17392231107@163.com (Y.S.); huanan.yu@csust.edu.cn (H.Y.); 3Henan Provincial Communications Planning & Design Institute Co., Ltd., Zhengzhou 450000, China; 13142277015@163.com; 4School of Materials Science and Engineering, Central South University, Changsha 410017, China; sy164798189@163.com

**Keywords:** road engineering, recycling, waste printed circuit board, asphalt material, rheological properties, micro-analysis

## Abstract

Non-metallic fractions (NMFs) from waste printed circuit boards (PCBs) are mostly composed of cured resin and fiber. In this study, NMF material from a PCB was ground into powder and added into matrix asphalt to produce PCB-NMF-modified asphalt. To improve the compatibility of PCB-NMF and asphalt, a compatibilizer consisting of tung oil and glycerol was also developed. The optimum compatibilizer content was determined to be 8% by weight of the PCB-NMF through a series of laboratory tests, including the softening point, penetration, ductility, and softening point difference (SPD). The micro-mechanism of NMF powder-modified asphalt was analyzed through Fourier transform infrared spectroscopy (FTIR) and a scanning electron microscope test (SEM). The performances of PCB-NMF-modified asphalt were evaluated by the dynamic shear rheology (DSR) test and the low-temperature bending beam rheometer (BBR) test. The optimum compatibilizer content was 8% by weight of the NMF powder and the optimum content of NMF powder was determined to be 30% by weight of the asphalt based on a comprehensive evaluation. The results show that PCB-NMF can significantly improve stiffness, rutting resistance, high-temperature stability, and temperature sensitivity of asphalt material at an appropriate content. The BBR tests revealed that PCB-NMF slightly weakened the cracking resistance of asphalt at low temperatures. The SEM test showed that the addition of a compatibilizer can increase the compatibility by making the NMF powder evenly dispersed. The FTIR test results implied that a chemical reaction may not have happened between PCB-NMF, compatibilizer, and the matrix asphalt. Overall, it is a promising and sustainable way to utilize PCB-NMF as a modifier for asphalt material and reduce electronic waste treatment at a low cost.

## 1. Introduction

Electronic products are of great use in people’s daily lives; this has resulted in rapid growth in the scrap rate of PCBs in the last few decades [1]. Globally, 53.6 million tons of e-waste were generated in 2019 and 74.7 million tons are expected by 2030 [2]. Currently, the most common type of way involves direct disposal via landfills or incineration, which contaminate the land and pollute the air, threatening the environment and human health [3]. Instead of landfills or incineration (directly), it might be a good option to use waste PCBs in highway construction. The application of solid waste in asphalt material has been a hot topic in academics and engineering. The authors of [4,5] added waste plastics to asphalt mixtures and evaluated the performance of the waste plastics–added mixes. Lakusic [6] added e-waste to asphalt mixtures as an aggregate replacement material, and the results showed that the waste plastic particles can be introduced in bituminous mixes as percentages of the weight replacements of coarse aggregate. Hayat et al. [7] attempted to add plastic waste into reclaimed asphalt pavement; their results showed that utilization of plastic waste in asphalt pavement enhanced the performance. In [8], de Almeida Júnior et al. used scrap tires instead of styrene–butadiene–styrene triblock copolymer (SBS) to prepare modified asphalt. Ilyin et al. [9] found that the heat resistance of asphalt was enhanced by adding SBS and activating the powder of discretely devulcanized rubber (APDDR). Lv et al. [10] explored the use of crayfish shells in the process of highway construction. Li et al. [11] developed crumble rubber tier-modified asphalt for the interlayer in a flexible-rigid composite pavement. Luhar and Luhar [12] investigated the possible application of e-waste in the building industry and found that doing so could minimize carbon emissions and resource usage.

A few studies have been undertaken in recent years to better address the issue of electronics-related waste in the road engineering field. E-waste plastic was employed in some studies to increase the performance of asphalt. Colbert and You [13] added e-waste powder made from recovered computer plastics into asphalt to increase the viscosity, blending temperatures, and rutting resistance of the binder. Mohd Hasan et al. [14] studied the performance of e-waste-modified asphalt binders, finding considerable increases in stiffness and elasticity after the e-waste polymers were treated with cumene hydroperoxide. Shahane and Bhosale [15] studied bitumen modified with 5% e-waste plastic powder, finding that the dynamic modulus increased by 10% at 40 °C, fatigue resistance increased by 19% at 10 °C, and rutting resistance increased by 28% at 40 °C. Ban et al. [16] employed waste PCB as a cement mortar addition, and the results showed that the compressive strength of waste PCB mortar is about equal to that of conventional cement mortar (and even slightly greater when used in an acidic environment). Many studies were conducted on the application of waste NMF in highway construction. Dong et al. [17] examined the effects of NMF powder content on the compatibility and microstructure of NMF-modified asphalt and discovered that as the NMF content increased, the compatibility and storage stability of PCB-modified asphalt deteriorated. According to the study, the optimal level of NMF in asphalt should be no more than 10% by weight. Guo et al. [18] studied the impacts of non-metal content and particle sizes on the high-temperature viscosity, penetration, softening point, ductility, and rheological properties of NMF powder-modified asphalt. The addition of NMF powder increased the stiffness and rutting resistance of the mixture but decreased its temperature susceptibility and ductility, with the modified asphalt with the smallest particle size of NMF powder having the highest performance in the penetration index, equivalent softening point, and equivalent brittle point. However, the study did not consider the impact of compatibility on NMF-modified asphalt. Yang et al. [19] modified asphalt with high boiling point fractions from waste PCB pyrolysis oil; it was discovered that when the modifier amount is less than 10%, the rutting resistance and moisture stability of the asphalt mixture are better, and the softening point of modified asphalt is higher. Yu et al. [20] tested modified asphalt with various PCB powder amounts; they discovered that adding PCBs improved the high-temperature performance, low-temperature performance, and temperature sensitivity of the asphalt material, but had a negative influence on the fatigue life. According to Meng et al. [21], the performance test of PCBs and styrene-butadiene rubber (SBR) composite-modified bituminous mix demonstrated that PCBs may greatly increase the rutting resistance and water stability of SBR-modified bitumen at high temperatures at the specified optimum concentration. At low temperatures, the crack resistance is diminished, yet it still meets the engineering criteria.

The two main components of NMF have cured epoxy resin and glass fiber. There are many studies on epoxy resin modified asphalt, and the application of epoxy resin in asphalt modification is well established, but there are almost no studies on the application of cured epoxy resin in asphalt mixes. However, there are some studies on the application of glass in asphalt mixes. Ghasemi and Marandi [22] studied the efficiency of bitumen and asphalt mixtures modified with crumb rubber and recycled glass powder, concluding that using recycled glass powder (RGP) instead of crumb rubber (CR) has no negative impact on the efficiency of bitumen and asphalt mixtures and improves their engineering properties, except for the toughness index. Androjić and Dimter [23] replaced some of the aggregates and fillers in the asphalt mix with crushed glass and found that increasing the percentage of glass reduced the density, stability, and void content of the mix. Jin et al. [24] used cathode-ray tube glass powders to modify asphalt, and the results revealed that cathode-ray tube (CRT) glass powders may be recycled in asphalt binders as an environmentally beneficial recycling technology, with a CRT glass powder addition percentage of up to 10%. (wt.). To improve the compatibility of polymer and asphalt of polymer-modified asphalt, several studies have been conducted. Nie et al. [25] used waste bio-oil as a softening and compatibility agent for high-content SBS-modified asphalt. Results showed that the addition of waste bio-oil improved the elasticity, flexibility, and anti-aging behaviors of the asphalt binder. Wang et al. [26] used tung oil, dioctyl phthalate (DOP), C9 petroleum resin, and organic montmorillonite (OMMT) to prepare the composite regenerating agent. Results showed that the addition of the tung oil composite regenerating agent can make the asphalt surface smooth, indicating that the tung oil composite regenerating agent can restore the microstructure and morphology of aged asphalt to a certain extent.

Proper use of non-metallic parts of waste printed circuit boards (PCB-NMF) in pavement engineering can, on the one hand, diminish the e-waste, and on the other hand, improve the properties of paving materials. This paper studies the possibility of recycling PCN-NMF in asphalt material, using tung oil and glycerin as composite compatibilizers. The mechanism of the compatibilizer was investigated, and the performance of PCB-NMF-modified asphalt was evaluated through a series of laboratory tests. The results can be used as a guide for preparing modified asphalt using scrap circuit boards.

## 2. Materials and Methodologies

### 2.1. Materials

The asphalt material was matrix asphalt no. 70 (asphalt with penetration 70) from Shell (Canton, China) Co., Ltd., the properties of which are listed in Table 1, and the technical requirements were from the Chinese standard test methods of bitumen and bituminous mixtures for highway engineering (JTGE20-2011) [27]. A non-metallic fraction (NMF) from a waste printed circuit board (PCB) (termed PCB-NMF) was composed of about 60% cured thermosetting epoxy resin and 40% glass fiber, with an average particle size of 23 μm.

PCB-NMF is a polymer composite material composed of mainly cured epoxy resin and some glass fiber. The properties of PCB-NMF are different from those of asphalt, resulting in storage instability. According to related research [28] and a previous study, a compatibilizer consisting of 60% tung oil and 40% glycerin was adopted to improve the compatibility of PCB-NMF and asphalt. As the main component, the properties of the compatibilizer are listed in Table 2.

### 2.2. Preparation Process of NMF-Modified Asphalt

The different contents of the compatibilizer were used to pretreat PCB-NMF, i.e., heated PCB-NMF was soaked in a compatibilizer and manually stirred for 5 min. After that, the PCB-NMF and compatibilizer mix were added to the matrix asphalt, and then PCB-NMF-modified asphalt was produced by the high-speed shear method, with a temperature of 175 °C ± 5 °C and a shear rate of 3500 r/min for 45 min. At last, the mixture was kept in the oven at 175 °C for 70 min to make the final product. The preparation process of PCB-NMF-modified asphalt is shown in Figure 1. Modified asphalt samples with 0%, 4%, 8%, 12%, and 16% compatibilizer by weight of PCB-NMF and 10%, 20%, 30%, and 40% PCB-NMF were prepared by this method for the following tests and evaluations.

### 2.3. Test Methods

All tests were conducted according to the test scheme shown in Figure 2.

(1) Scanning electron microscopy.

To further confirm the influence of the compatibilizer on PCB-NMF and the matrix asphalt, the samples with 0%, 4%, 8%, 12%, and 16% compatibilizer by weight of PCB-NMF were conducted with an SEM test. The 81W/AIS2100 electron microscope was adopted in this test. 

(2) Rotational viscosity (RV).

The viscosity is an important technical indicator for the workability of the asphalt material. The DV2T Brookfield rotational viscometer was utilized in this study to evaluate the viscosity of the PCB-NMF-modified asphalt samples at 120 °C, 135 °C, and 175 °C. Three replicates were fabricated and tested to obtain reliable results.

(3) Storage stability.

To evaluate the storage stability of PCB-NMF-modified asphalt, the phase separation test was applied. A total of 50 g of heated asphalt binders were poured into a sample tube (which was then placed vertically in an oven for 48 h at 165 °C). After they were taken out and rested until they cooled to room temperature, the sample tubes were divided into three equal parts. The difference in the softening point between the upper and lower parts of the sample tube was calculated, as in Equation (1), to evaluate the storage stability of PCB-NMF-modified asphalt. The SPD is the ratio of the penetration of the upper segment and lower segment of tubed samples after being vertically stored for 24 h, calculated by Equation (1).
SPD=|S_u_ − S_l_|(1)
where: S_u_ is the softening point of the upper section of the sample at °C. S_l_ is the softening point of the lower section of the sample at °C.

(4) Dynamic shear tests.

Temperature sweep tests were conducted using the dynamic shear rheometer model Anton Paar Smart Pave 102 to study the dynamic shear rheometric properties of PCB-NMF-modified asphalt materials. The asphalt specimens were sandwiched between two 25 mm diameter parallel plates with a gap of 1 mm, one of which was fixed, and the other revolved around the central steel axis. A sinusoidal shear load was implemented to the sample at an angular frequency of 10 rad/s at the strain control mode. The temperature sweep was arranged from 40 °C to 90 °C.

(5) Bending beam rheology.

To test the low-temperature performance of PCB-NMF-modified asphalt material, the asphalt samples were tested using a bending beam rheometer at a decreasing temperature gradient: −12 °C, −18 °C, and −24 °C. 

(6) Fourier transform infrared spectroscopy.

To further understand the modification mechanism of PCB-NMF on asphalt, the interaction effects between PCB-NMF and virgin asphalt were investigated by using a Fourier transform infrared spectroscopy. The samples were dissolved in a prepared carbon tetrachloride solution, and then each sample was put on a potassium bromide (KBr) tablet for testing. The wavenumber range was from 500 cm^−1^ to 4000 cm^−1^, the resolution was 1~0.4 cm^−1^, the wavenumber accuracy was 0.01 cm^−1^/2000 cm^−1^, and the transmittance accuracy was 0.01%.

## 3. Results and Discussions

### 3.1. Compatibilizer Content Affection Analysis

#### 3.1.1. Determination of Optimum Compatibilizer Content

To determine the optimum compatibilizer content, modified asphalt with 20% PCB-NMF was treated with five compatibilizer contents: 0%, 4%, 8%, 12%, and 16% by weight of PCB-NMF, and then sheared with matrix asphalt to make PCB-NMF asphalt samples for the test. The optimum compatibilizer content was determined by the softening point, penetration, ductility, and SPD. 

As shown in Figure 3a–d, the softening point of PCB-NMF-modified asphalt increased first with the increase of the compatibilizer and then decreased after reaching the peak, while penetration and SPD showed an opposite trend. The softening point increased when the compatibilizer was below 8%, and increased by 4.6%. At the same time, penetration and SPD decreased by about 6.5% and 17%, respectively. However, when the compatibilizer further increased from 8% to 16%, the softening point decreased by about 8%, while penetration and SPD increased by 20% and 15%, respectively. Ductility—different from the softening point—constantly increased until the content of the compatibilizer reached 12% (increasing by 188%). Moreover, the compatibilizer increased from 12% to 16%, the ductility decreased by 50%. This is because excessive tung oil would actually soften the asphalt material and increase the fluidity of asphalt, bringing adverse effects to the properties of asphalt. Overall, when the compatibilizer is 8%, except for ductility, the properties of PCB-NMF asphalt are the best. Thus, 8% tung oil by weight of PCB-NMF is determined to be the optimum compatibilizer content.

#### 3.1.2. SEM Analysis

To describe the micro-scale state of PCB-NMF-modified asphalt, before and after crushing, as well as how the compatibilizer improves the compatibility between PCB-NMF and asphalt, magnified images (600× and 800×) of asphalt with different PCB-NMF content were obtained from an electron microscope. The images at 0%, 4%, 8%, 12%, and 16% of the compatibilizer by weight of PCB-NMF are shown in Figure 4.

It is clearly shown that there is a microstate change of the PCB-NMF particles in the asphalt material under the influence of the compatibilizer. After the PCB-NMF powder is added to the matrix material, due to the difference in gravity and polarity, they cannot form a stable system. As a result, PCB-NMF powder gathers and forms clusters (as marked in the red circle in Figure 4a), which cause non-homogeneity. With the compatibilizer added, the PCB-NMF particle size significantly decreases, as Figure 4b–d shows. When the compatibilizer was 8%, the PCB-NMF was evenly dispersed in the asphalt material.

### 3.2. Basic Properties Analysis

The penetration, softening point, and ductility of PCB-NMF-modified asphalt are shown in Figure 5a–c. It can be seen that the penetration and ductility decreased with the increase of PCB-NMF powder content. When the amount of PCB-NMF powder was 10%, 20%, 30%, and 40%, the penetrations decreased by 21%, 34%, 52%, and 71%, respectively, and the ductility decreased by 14%, 22%, 35%, and 78%, respectively. On the other hand, the softening point increased with the increase of PCB-NMF powder content. When the PCB-NMF powder was 10%, 20%, 30%, and 40%, the softening point increased by 11%, 24%, 27%, and 31%, respectively.

The viscosities of the asphalt samples with different NMF contents are shown in Figure 6a. The viscosity–temperature index (VTS) of each asphalt at the high-temperature stage (135 °C to 175 °C) is calculated by Equation (2) and displayed in Figure 6b.
(2)$$\mathrm{VTS}=\frac{\mathrm{lg}\mathrm{lg}(\eta_{1}\times 10^{3})-\mathrm{lg}\mathrm{lg}(\eta_{2}\times 10^{3})}{\mathrm{lg}(T_{1}+273.13)-\mathrm{lg}(T_{2}+273.13)}$$
where: *η*_1_ and *η*_2_ are the viscosities at 135 °C and 175 °C in Pa·s. *T*_1_ and *T*_2_ are the different test temperatures for asphalt viscosities at °C.

It is clearly shown that the PCB-NMF powder significantly increased the viscosity of PCB-NMF-modified asphalt. When the PCB-NMF powder was 10%, 20%, 30%, and 40%, the viscosities at 135 °C were 44%, 92%, 297%, and 475% higher than the control sample. In accordance with the Superpave specifications, the rotational viscosity of the asphalt binder at 135 °C should be less than 3 Pa·s to ensure workability. As shown in Figure 6a, the viscosities of all samples, except the 40% PCB-NMF-modified asphalt, meet this requirement. The viscosity–temperature index characterizes the temperature sensitivity of asphalt material: the smaller the absolute value of VTS, the less temperature sensitivity of the asphalt material. It is notable from Figure 6 that the absolute values of the VTS are in decreasing order with PCB-NMF content. The preliminary results revealed that the existence of PCB-NMF powder improved the high-temperature rheological properties of the asphalt binder (regarding high-temperature stability and temperature sensitivity). The enhancements were mainly due to the volumetric filling of the powder particles. Moreover, the PCB-NMF powder adsorbed the light component of the asphalt and formed a more stable blend.

### 3.3. Storage Stabilities

To evaluate the storage stability of PCB-NMF-modified asphalt, the SPD was calculated through the softening point difference of the upper and lower sections of the sample tube, as Equation (1). Table 3 shows the high-temperature storage stability of asphalt modified with different PCB-NMF contents. Generally speaking, the higher the SPD, the more severe the phase separation of the asphalt binder. Overall, the SPD increases with PCB-NMF content. This is because as PCB-NMF content increases, more PCB-NMF particles are deposited to the bottom of the asphalt binder due to gravity, making the lower part of the asphalt samples harder. According to the specification, if the SPD of the asphalt binder is higher than 2.2 °C, it would be considered storage unstable. In this regard, the SPDs of PCB-NMF-10, PCB-NMF-20, and PCB-NMF-30 are 0.8 °C, 1.4 °C, and 1.9 °C, respectively. However, the SPD of PCB-NMF-40 is more than 2.2 °C. As a result, the percentage of PCB-NMF powder should be controlled under 40% according to this study, achieving acceptable storage stability.

### 3.4. Rheology Analysis

The complex modulus G* and phase angle δ are shown in Figure 7, and the rutting factor G*/sinδ is shown in Figure 8. 

In Figure 7a, the G* of all samples decreased as the sweep temperature increased. According to the study by Ilyin and Strelets [29], this phenomenon is due to the loss of the glass state of the asphalt binder. Smaller G* means less resistance to shear deformation. From Figure 7, the G* increased with the increasing content of PCB-NMF powder. For instance, compared with that of the control asphalt binder, the average G* values of modified asphalt with 30% PCB-NMF powder at 46 °C, 52 °C, 58 °C, 64 °C, 70 °C, and 76 °C increased by 7835 Pa, 4650 Pa, 1735 Pa, 880 Pa, 577 Pa, and 325 Pa, corresponding to increase magnitudes of 58%, 84%, 71%, 82%, 114%, and 127%, respectively. These indicated that the addition of PCB-NMF could significantly increase the shear deformation resistance of asphalt material. Nevertheless, the increase of PCB-NMF powder on G* was at a decreasing rate. For example, when the PCB-NMF content increased from 0 to 10%, the increase of G* at 64 °C was 32%. Comparatively, when the PCB-NMF further increased from 30% to 40%, the increase of G* was down to 19%.

The proportion of elasticity to the viscosity of asphalt material can be characterized by phase angle δ: tan δ = G″/G′. The smaller the δ is, the more elastic the asphalt material. From Figure 7b, the δ of all asphalt materials increased with the temperature. More importantly, PCB-NMF-modified asphalt demonstrated smaller δ as the content of PCB-NMF powder increased for a given temperature. Compared to the control asphalt, the average δ values of modified asphalt with 30% PCB-NMF powder at 46 °C, 52 °C, 58 °C, 64 °C, 70 °C, and 76 °C were 1.1°, 1.6°, 0.9°, 0.7°, 0.1° lower than control asphalt binder, corresponding to decrease magnitudes of 1.44%, 1.88%, 1.04%, 0.80%, 0.11%, and 0.45%, respectively. The reduction of δ indicates that the addition of PCB-NMF powder can increase the deformation resistance of asphalt material, improving its resistance and resilience to permanent deformation. This is due to the molecule interaction of PCB-NMF and asphalt that improves its elasticity.

The rutting factor |G*|/sinδ was proposed to estimate the high-temperature performance of the asphalt binder in the strategic highway research program (SHRP) criteria. In general, a larger rutting factor indicates better rutting resistance. It can be seen from Figure 8 that the rutting factor of the asphalt binder decreased as the temperature increased. For a given temperature, the rutting factor |G*|/sinδ of PCB-NMF powder-modified asphalt was always larger than the control asphalt. Compared with the control group, the average G* values of asphalt modified with 30% PCB-NMF powder at 46 °C, 52 °C, 58 °C, 64 °C, 70 °C, and 76 °C increased by 7952 Pa, 4698 Pa, 1743 Pa, 882 Pa, 577 Pa, and 325 Pa, corresponding to increased magnitudes of 58%, 84%, 71%, 82%, 114%, and 127%, respectively. Similarly, the rutting factor |G*|/sinδ of 10%, 20%, 30%, and 40% PCB-NMF as well as the control asphalt met the SHRP requirement (>1.0 kPa) at 64 °C. However, when the temperature reached 70 °C, the rutting factors of the control asphalt and asphalt samples modified with 0%, 10%, 20%, 30%, and 40% of PCB-NMF were 0.505 kPa, 0.836 kPa, 0.953 kPa, 1.082 kPa, and 1.190 kPa, respectively. Only the rutting factor of 30% and 40% PCB-NMF-modified asphalt still met this requirement (>1.0 kPa). The results showed that the high-temperature performance of asphalt was improved by the PCB-NMF powder. By comprehensively checking the effects of PCB-NMF viscosity and dynamic shear properties, the optimum content of PCB-NMF was determined as 30% by the weight of the asphalt material. According to Table 4, the high-temperature grade of performance grade (PG) classification of 30% PCB-NMF asphalt was increased.

### 3.5. Low-Temperature Performance

Two parameters: creep stiffness (S) and change rate of creep stiffness (m-value) were used in the BBR test to evaluate the low-temperature properties of the PCB-NMF-modified asphalt. The binder with a higher m-value and lower creep stiffness possesses better thermal cracking resistance at low temperatures.

In Figure 9, one can see that compared to the matrix asphalt, the creep stiffness of PCB-NMF-modified asphalt was higher, while the m-value was lower. Moreover, as the content of PCB-NMF increased, the creep stiffness of the asphalt climbed, but the m-value dropped, indicating that PCB-NMF may have an adverse effect on the low-temperature properties of asphalt. The addition of PCB-NMF hardens the asphalt and increases the chances of low-temperature cracking. According to the specification AASHTO T 313, the asphalt’s creep stiffness should be lower than 300 MPa, and the m-value should be higher than 0.3. The test results show that the creep stiffness and m-value of PCB-NMF passed the low-temperature criteria at −12 °C but failed at −18 °C. In this sense, PCB-NMF is not able to enhance the thermal cracking resistance at low temperatures.

### 3.6. FTIR Analysis

The FTIR analysis results of both the matrix asphalt and PCB-NMF-modified asphalt are shown in Figure 10. The infrared spectra of the matrix asphalt and PCB-NMF asphalt are similar. The two absorption peaks in the range of 2500–3000 cm^−1^ are caused by the C–H vibrational contraction in the aliphatic chain because of the saturated hydrocarbon of the asphalt. The wavelength in the range of 1500–1750 cm-1 was correlated to the conjugated double bond of C=C and C=O due to the aromatic compounds of the asphalt; –CH_2_ and –CH_3_ in wavelength ranges of 1250–1500 cm^−1^ were caused by the internal bending vibration of the C–H plane, which can prove the existence of alkanes in the asphalt. The absorption peak appears in the wavelength range of 750–1000 cm^−1^ and is correlated to unsaturated C–H and C–C skeleton vibrations in the asphalt.

A comparison of the matrix asphalt and PCB-NMF asphalt shows some slight changes in the intensities of absorption peaks, with two weakening effects of the absorption peak: one is the contractional vibration of the saturated hydrocarbon (C–H) at 2750 cm^−1^, and the other is the absorption peak of the internal bending vibration of –CH_2_ and –CH_3_ at around 1500 cm^−1^. A chemical reaction may not have happened between PCB-NMF, compatibilizer, and the matrix asphalt.

## 4. Conclusions

This study tackles the environmental burdens caused by the electronic waste of printed circuit boards and enhances the mechanical performance of asphalt binders. The major novelty and scientific contributions of this study can be summarized as follows:(1)A compatibilizer consisting of tung oil and glycerin can help disperse the PCB-NMF powder particles in asphalt materials, improving the compatibility of PCB-NMF powder and asphalt. The optimum compatibilizer content is determined as 8% by the weight of PCB-NMF.(2)Through a series of tests, it was found that the addition of PCB-NMF can increase the dynamic modulus, viscosity, and softening point. The appropriate content of PCB-NMF significantly improves the stiffness, rutting resistance, and temperature sensitivity of asphalt materials.(3)By comprehensively checking the effects of PCB-NMF powder on penetration, softening point, ductility viscosity, and dynamic shear properties of modified asphalt, the optimum content of PCB-NMF is determined as 30% by the weight of the asphalt.(4)The BBR test demonstrates that PCB-NMF increases the creep stiffness and decreases the m-value of the binder. Although the asphalt modified with 30% PCB-NMF meets the low-temperature criterion at −12 °C, the low-temperature performance is weakened.

Overall, this study provides insights into innovatively-recycling non-metallic fractions of printed circuit boards in pavement engineering. It is expected that using PCB-NMF as a modifier will not only improve the properties (especially high-temperature properties) of asphalt material but also potentially reduce electronic waste and ease environmental pollution.

## Figures and Tables

**Figure 1 materials-15-04172-f001:**
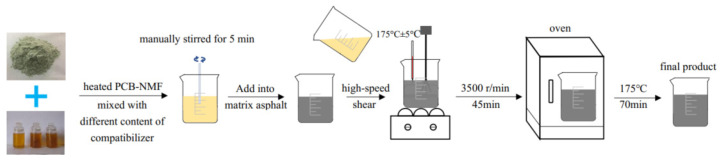
PCB-NMF asphalt preparation process.

**Figure 2 materials-15-04172-f002:**
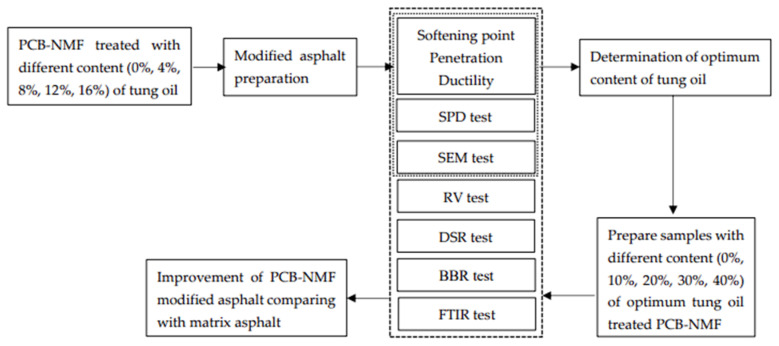
PCB-NMF-modified asphalt test scheme.

**Figure 3 materials-15-04172-f003:**
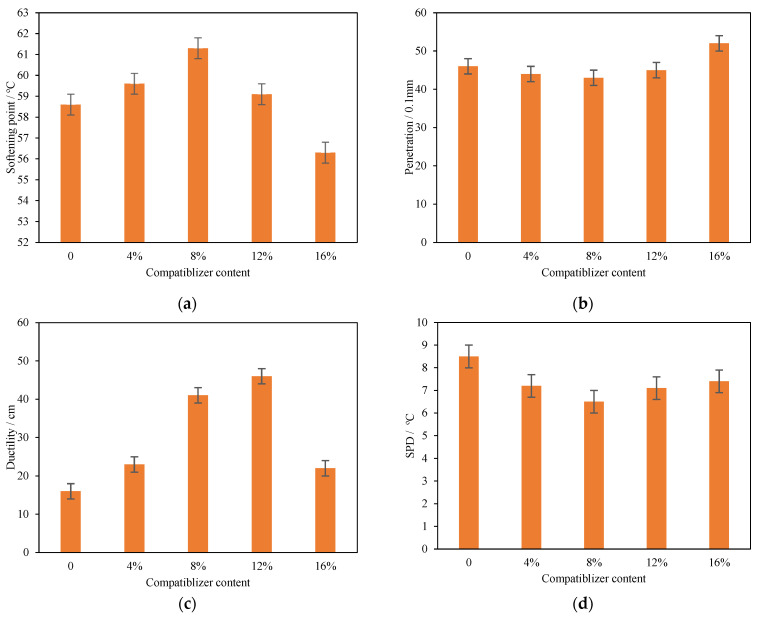
Properties of the 20% PCB-NMF-modified asphalt with different compatibilizer contents: (**a**) softening point; (**b**) penetration; (**c**) ductility; (**d**) SPD.

**Figure 4 materials-15-04172-f004:**
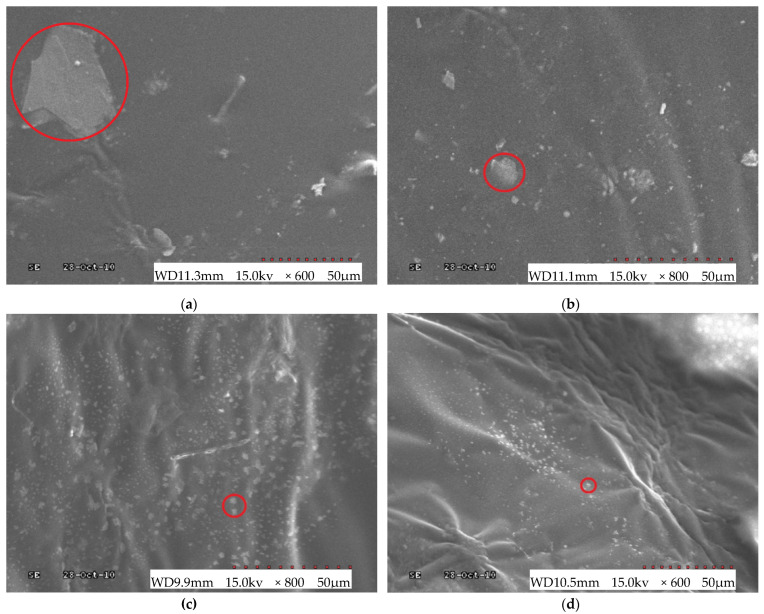
SEM of NMF modified asphalt: (**a**) 0% compatibilizer content (magnitude ×600); (**b**) 4% compatibilizer content (magnitude ×800); (**c**) 8% compatibilizer content (magnitude ×800); (**d**) 12% compatibilizer content (magnitude ×600).

**Figure 5 materials-15-04172-f005:**
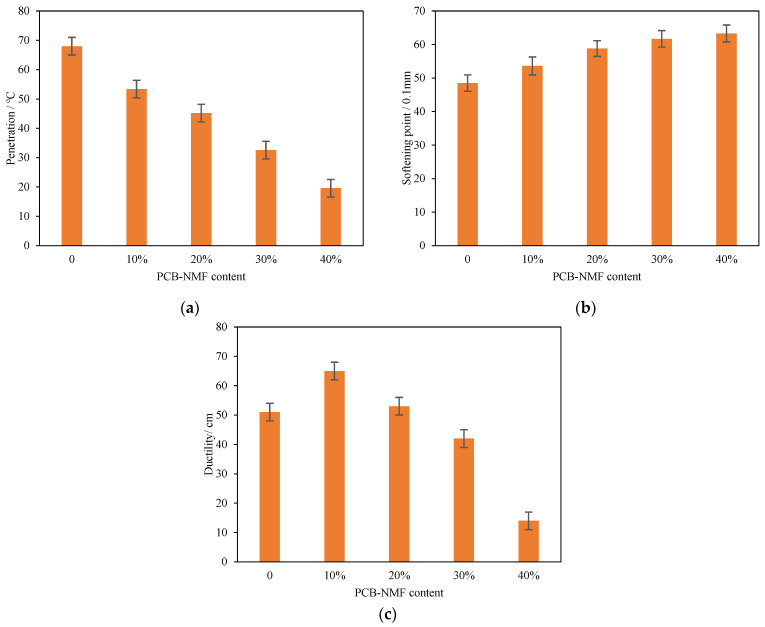
0%, 10%, 20% 30%, and 40% of PCB-modified asphalt tested for: (**a**) penetration, (**b**) softening point, and (**c**) ductility.

**Figure 6 materials-15-04172-f006:**
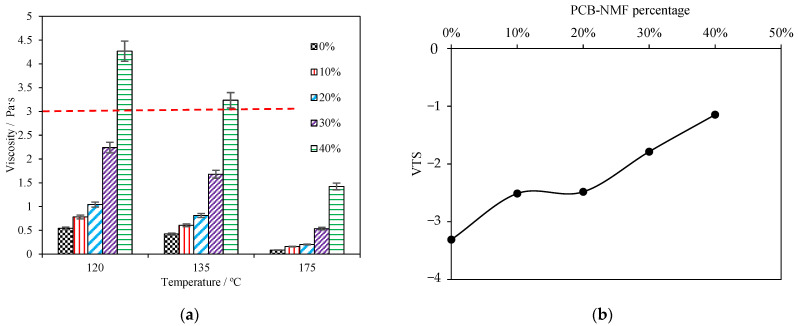
Viscosity and VTS of asphalt with different PCB-NMF contents: (**a**) viscosity, (**b**) VTS.

**Figure 7 materials-15-04172-f007:**
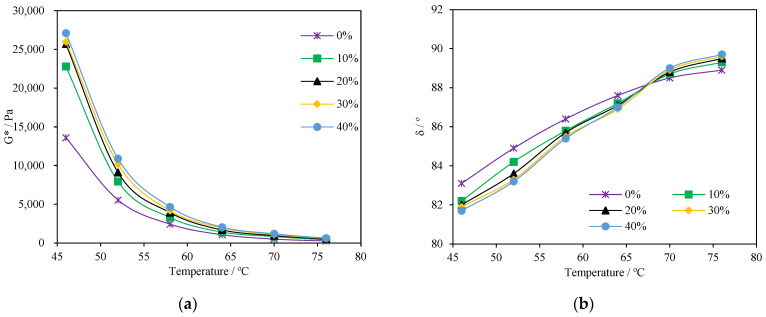
Dynamic rheology properties of asphalt with different PCB-NMF content: (**a**) dynamic shear module G*; (**b**) phase angle δ.

**Figure 8 materials-15-04172-f008:**
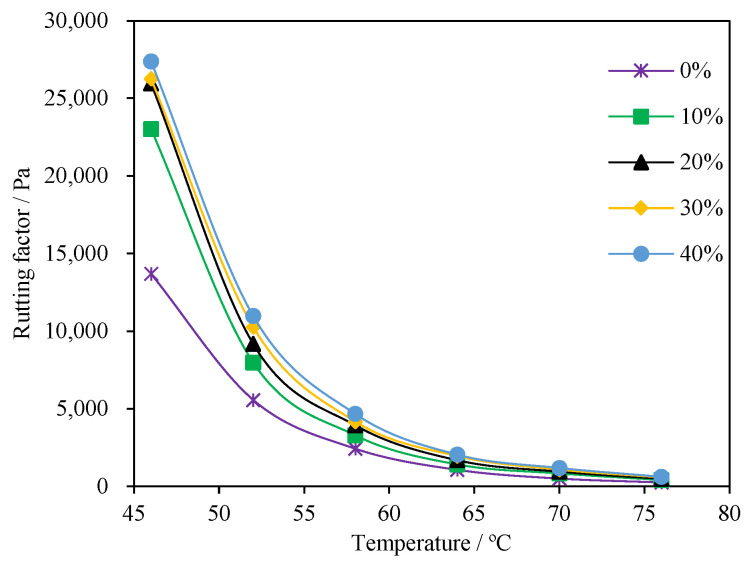
Effect of PCB-NMF content on the asphalt rutting factor G*/sinδ.

**Figure 9 materials-15-04172-f009:**
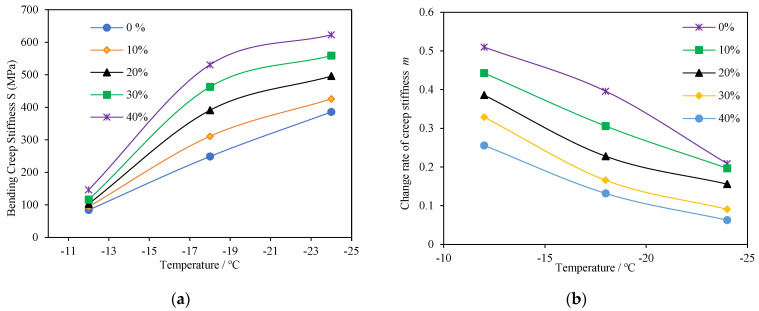
Low-temperature properties of asphalt with different PCB-NMF contents; (**a**) creep stiffness; (**b**) change rate of creep stiffness.

**Figure 10 materials-15-04172-f010:**
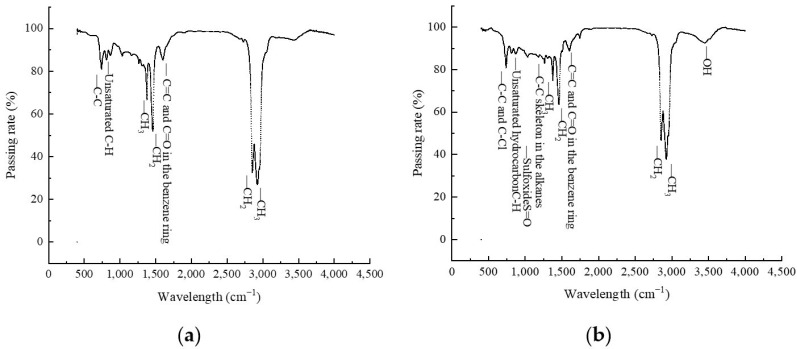
The infrared spectrum of matrix asphalt and PCB-NMF-modified asphalt: (**a**) matrix asphalt; (**b**) PCB-NMF (30%) modified asphalt.

**Table 1 materials-15-04172-t001:** Main technical indexes of matrix asphalt no. 70.

Test Indicators	Unit	Technical Requirements	Test Result
Penetration 25 °C, 100 g, 5 s	0.1 mm	60~70	68
Softening point	°C	≥47	48.5
Ductility 10 °C, 5 cm/min	cm	≥20	51
Residual material loss after RTFOT *	%	≤±0.8	0.1
Penetration ratio of residue after RTFOT * (25 °C)	%	≥61	70.0

* RTFOT means rolling thin film oven test.

**Table 2 materials-15-04172-t002:** The physical and chemical properties of compatibilizer.

Test Indicators	Test Result
Exterior	Brown–yellow liquid
Odor	Odor of tung oil
Transparency (24 h/20 °C)	A small amount of precipitation
Relative density (20 °C)	0.9360–0.9395
Refractive index (20 °C)	1.5170–1.5220
Iodine number	163–173
Saponification number	190–195
Drying time	3–7 days in summer, 3–20 days in winter
Acid value (mgKOH/g)	<7
Moisture and volatile matter content	<0.20%
Impurity	<0.20%
β-tung oil test (3.3 °C–4.4 °C for 24 h)	No crystallization

**Table 3 materials-15-04172-t003:** SPD of asphalt with different PCB-NMF contents.

PCB-NMF percentage (%)	0	10	20	30	40
SPD (°C)	0	0.8	1.4	1.9	2.4

**Table 4 materials-15-04172-t004:** High-temperature grade of PG classification.

Item		Result	Requirement
Matrix asphalt	Dynamic cut 64 °C, G^*^/sinδ, kPa	1.07	≥1.0
	Dynamic cut 70 °C, G^*^/sinδ, kPa	0.50	
30% NMF powder modified asphalt	Dynamic cut 70 °C, G^*^/sinδ, kPa	1.08	
	Dynamic cut 76 °C, G^*^/sinδ, kPa	0.58	

## Data Availability

The data used to support the findings of this study are available from the corresponding author upon request.

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
