# Peer review of "Recycling Non-Metallic Powder of Waste Printed Circuit Boards to Improve the Performance of Asphalt Material"

_materials, 2022, doi:10.3390/ma15124172_

Round 1
Reviewer 1 Report
The article by Li S. et al. discusses the modification of asphalt binder using waste epoxy printed circuit boards and a stabilizing oil. The authors consider how the introduction of these additives changes the rheological characteristics of the binder and some of its operational properties. In general, this article can be published in Materials but only after major revision. In addition, careful English editing of the text is highly desirable.
The specific comments are as follows.
Lines 19, 128: Here should be “cured resin” instead of “resin”. Resin is a liquid.
Keywords: Here (and everywhere else) should be “rheological properties” instead of “rheological property”. There are many kinds of rheological properties.
Graphic abstract: “Conventional property”, “Dynamic rheology property”, “High temperature property” and “Low temperature property”. These are not properties. There are characteristics such as stiffness, viscosity, storage modulus, softening point, and so on. In addition, there are properties such as shear resistance, rutting resistance, and so on.
Lines 55, 57: “de Almeida Júnior et al. [7] used scrap tires instead of SBS to prepare modified asphalt” and “Li et al. [9] developed crumble rubber tyer modified asphalt”. The use of crumble tires in the form of devulcanized rubber for the preparation of modified bitumen and a comparison with the use of SBS has also been discussed in another article (doi 10.1134/S1061933X1404005X).
Line 79: “classical properties”. Please replace this phrase with an indication of specific properties or characteristics. There is no such thing as classical properties.
Line 83: “PI, T800, and T1.2”. Please decipher these abbreviations.
Line 91: Here should be “[19] the” instead of “[19] The”.
Lines 56 and 91: Decipher these abbreviations: SBS and SBR.
Lines 103, 108: Decipher the abbreviations RGP, CR and CRT.
Line 112: Here should be “can on the one hand diminish the e-waste and on the other hand improve the properties of paving materials” instead of “can one hand dimimish the e-waste, on anther hand improve the property of paving mateirals”.
Line 114: The introduction does not say anything about tung oil and glycerin. Why are they chosen? Why is glycerin needed?
Lines 114-116: “The mechniasm…” Here is a record sentence in terms of the number of spelling and grammatical errors.
Table 1. The abbreviation RTFOT must be deciphered in a note to the table.
Figures 1 and 2. These figures should be deleted as having no useful meaning.
Lines 129, 191. The dielectric constant has no relation with storage instability. It should be removed from the text.
Line 142: Here should be “kept” or “held” instead of “developed”.
Line 144: Here it is necessary to write what specific mixtures have been prepared.
Lines 169-174: “This is due to the addition-esterification reaction between tung oleic acid produced by hydrolysis of tung oil and the epoxy group of the epoxy resin…” This cannot be the case since there are no epoxy groups in the cured epoxy polymer. This speculation must be removed.
Line 190: “that the microstate” -> “that there is microstate”.
Figure 6. The figures should have a scale on them.
Line 197: “and nearly fully dissolved” This should be removed. Cross-linked epoxy resin cannot dissolve.
Lines 197-202: “This is because… powder in the aggregate PCB-NMF powder”. This speculation should be removed. The cured resin has no reactive groups and cannot dissolve.
Line 220: “rhetoric” -> “rheometric”.
Figures 7 and 8. These figures should be deleted as being of no value.
Line 224: “a shear rate” -> “an angular frequency”.
Lines 229: “were measured” -> “were tested”.
Lines 239-240: “cm-1” -1 has to be a superscript.
Lines 149-203. This part should be moved into Results and discussions.
Everywhere: “declined” -> “decreased”.
Line 260: “asphalt for:” -> “asphalt tested for:”
Line 275: “sensitivity the asphalt” -> “sensitivity of the asphalt”.
Line 281: absorbed -> adsorbed.
Line 282: a more stable sol-gel colloid -> a more stable blend.
Lines 299-300: Fig. 8 -> Fig. 11; Fig. 11 -> Fig. 12.
Line 301-305: These are common knowledge things that should be removed.
Line 307: “This is because asphalt binder is a typical viscous-elastic material. It changes from elastic state into non-Newtonian fluid state as temperature increases.” This sentence should be removed. In this case, G* is not decreasing for this reason. Its reduction is due to the loss of the glass state of asphalt binder (see 10.1021/acs.energyfuels.7b03058 for basic knowledge).
Line 329: asphalt improves -> asphalt that improves.
Line 331: property -> properties.
Line 334: Please decode the abbreviation SHRP.
Line 338: the average G* values -> the average G*/sinδ values?
Table 5. Title. Please decode the abbreviation PG.
Lines 391-393: “The changing of absorption peak intensity and emerge of new absorption peaks indicate chemical reactions between PCB-NMF, compatibilizer and matrix asphalt.” No, it does not. To prove this, the change in spectra during temperature treatment should be compared. This sentence should be removed as speculation.
Lines 395-398. This is speculation. It should be removed. Carbonyl groups cannot react with epoxy groups, which are also not present in the crosslinked resin.
Lines 399-412. This part should be removed because the DSC curves are not taken correctly. They should be obtained from low negative temperatures. Those that the authors show are not peaks caused by transitions but an instrumental error due to the calorimeter going into the measurement mode.
Lines 429-432. This should be removed as nothing more than pure speculation.
Author Response
Response to Reviewer 1 Comments
Dear Reviewers:
Thank you for your letter and for the reviewers’ comments concerning our manuscript entitled “Recycling Non-Metallic Powder of Waste Printed Circuit Board in Improving the Performance of Asphalt Material”. Those comments are all valuable and very helpful for revising and improving our paper, as well as the important guiding significance to our researches. We have studied comments carefully and have made correction which we hope meet with approval. The main corrections in the paper and the responds to the reviewer’s comments are as following:
Responds to the reviewer’s comments:
Point 1: Response to comment: Lines 19, 128: Here should be “cured resin” instead of “resin”. Resin is a liquid.
Response 1: Many thanks for this comment. All “resin” in the manuscript have been corrected to “cured resin”.
Point 2: Response to comment: Keywords: Here (and everywhere else) should be “rheological properties” instead of “rheological property”. There are many kinds of rheological properties.
Response 2: Many thanks for this comment. All “rheological property” in the manuscript have been corrected to “rheological properties”.
Point 3: Graphic abstract: “Conventional property”, “Dynamic rheology property”, “High temperature property” and “Low temperature property”. These are not properties. There are characteristics such as stiffness, viscosity, storage modulus, softening point, and so on. In addition, there are properties such as shear resistance, rutting resistance, and so on.
Response 3: Many thanks for this comment. It was decided to delete Graphic abstract because its representation was not clear enough and Figures 1 and 2 can also illustrates the entire flow of the study.
Point 4: Lines 55, 57: “de Almeida Júnior et al. [7] used scrap tires instead of SBS to prepare modified asphalt” and “Li et al. [9] developed crumble rubber tyer modified asphalt”. The use of crumble tires in the form of devulcanized rubber for the preparation of modified bitumen and a comparison with the use of SBS has also been discussed in another article (doi 10.1134/S1061933X1404005X).
Response 4: Many thanks for this comment. The article has been referenced in this manuscript.
Point 5: Line 79: “classical properties”. Please replace this phrase with an indication of specific properties or characteristics. There is no such thing as classical properties.
Response 5: Many thanks for this comment. The “classical properties” is the original statement of the article referenced here. The “classical properties” may refer to “High temperature viscosity”, “Penetration”, “Softening point” and “Ductility” according to the article. They have been listed one by one in the manuscript.
Point 6: Line 83: “PI, T800, and T1.2”. Please decipher these abbreviations.
Response 6: Many thanks for this comment. PI means penetration index, T800 means equivalent softening point, T1.2 means equivalent brittle point. These have been corrected in the article.
Point 7: Line 91: Here should be “[19] the” instead of “[19] The”.
Response 7: Many thanks for this comment. We have made correction according to the Reviewer’s comments.
Point 8: Lines 56 and 91: Decipher these abbreviations: SBS and SBR.
Response 8: Many thanks for this comment. We have made corrections in the article. SBS correct to butadiene–styrene–butadiene triblock copolymer, SBR correct to styrene-butadiene rubber.
Point 9: Lines 103, 108: Decipher the abbreviations RGP, CR and CRT.
Response 9: Many thanks for this comment. We have made corrections in the article. RGP correct to recycled glass powder, CR correct to crumb rubber, CRT correct to cathode-ray-tube.
Point 10: Line 112: Here should be “can on the one hand diminish the e-waste and on the other hand improve the properties of paving materials” instead of “can one hand dimimish the e-waste, on anther hand improve the property of paving mateirals”.
Response 10: Many thanks for this comment. We have made correction according to the Reviewer’s comments.
Point 11: Line 114: The introduction does not say anything about tung oil and glycerin. Why are they chosen? Why is glycerin needed?
Response 11: Many thanks for this comment. We have added relevant references and the reasons for using this compound compatibilizer.
Point 12: Lines 114-116: “The mechniasm…” Here is a record sentence in terms of the number of spelling and grammatical errors.
Response 12: Many thanks for this comment. We have corrected all the errors in this sentence.
Point 13: Table 1. The abbreviation RTFOT must be deciphered in a note to the table.
Response 13: Many thanks for this comment. We have added a note under the table.
Point 14: Figures 1 and 2. These figures should be deleted as having no useful meaning.
Response 14: Many Figures 1 and 2. These figures should be deleted as having no useful meaning.
Point 15: Lines 129, 191. The dielectric constant has no relation with storage instability. It should be removed from the text.
Response 15: Many thanks for this comment. We have removed it.
Point 16: Line 142: Here should be “kept” or “held” instead of “developed”.
Response 16: Many thanks for this comment. We have made correction according to the Reviewer’s comments.
Point 17: Here it is necessary to write what specific mixtures have been prepared.
Response 17: Many thanks for this comment. We have made correction according to the Reviewer’s comments.
Point 18: Lines 169-174: “This is due to the addition-esterification reaction between tung oleic acid produced by hydrolysis of tung oil and the epoxy group of the epoxy resin…” This cannot be the case since there are no epoxy groups in the cured epoxy polymer. This speculation must be removed.
Response 18: Many thanks for this comment. We have removed the speculation.
Point 19: Line 190: “that the microstate” -> “that there is microstate”.
Response 19: Many thanks for this comment. We have made correction according to the Reviewer’s comments.
Point 20: Figure 6. The figures should have a scale on them.
Response 20: Many thanks for this comment. We have made correction according to the Reviewer’s comments.
Point 21: Line 197: “and nearly fully dissolved” This should be removed. Cross-linked epoxy resin cannot dissolve.
Response 21: Many thanks for this comment. We have made correction according to the Reviewer’s comments.
Point 22: Lines 197-202: “This is because… powder in the aggregate PCB-NMF powder”. This speculation should be removed. The cured resin has no reactive groups and cannot dissolve.
Response 22: Many thanks for this comment. We have removed the speculation.
Point 23: Line 220: “rhetoric” -> “rheometric”.
Response 23: Many thanks for this comment. We have corrected it. Rhetoric corrected to rheometric.
Point 24: Figures 7 and 8. These figures should be deleted as being of no value.
Response 24: Many thanks for this comment. We have deleted them according to the Reviewer’s comments.
Point 25: Line 224: “a shear rate” -> “an angular frequency”.
Response 25: Many thanks for this comment. We have made a correction according to the Reviewer’s comments.
Point 26: Lines 229: “were measured” -> “were tested”.
Response 26: Many thanks for this comment. We have made correction according to the Reviewer’s comments.
Point 27: Lines 239-240: “cm-1” -1 has to be a superscript.
Response 27: Many thanks for this comment. We have made a correction according to the Reviewer’s comments.
Point 28: Lines 149-203. This part should be moved into Results and discussions.
Response 28: Many thanks for this comment. We have made an adjustment according to this comment.
Point 29: Everywhere: “declined” -> “decreased”.
Response 29: Many thanks for this comment. We have made correction according to the Reviewer’s comments.
Point 30: Line 260: “asphalt for:” -> “asphalt tested for:”.
Response 30: Many thanks for this comment. We have made a correction according to the Reviewer’s comments.
Point 31: Line 275: “sensitivity the asphalt”-> “sensitivity of the asphalt”.
Response 31: Many thanks for this comment. We have corrected it according to the Reviewer’s comments.
Point 32: Line 281: absorbed -> adsorbed.
Response 32: Many thanks for this comment. We have made a correction according to the Reviewer’s comments.
Point 33: Line 282: a more stable sol-gel colloid -> a more stable blend.
Response 33: Many thanks for this comment. We have made a correction according to the Reviewer’s comments.
Point 34: Lines 299-300: Fig. 8 -> Fig. 11; Fig. 11 -> Fig. 12.
Response 34: Many thanks for this comment. We have made a correction according to the Reviewer’s comments.
Point 35: Line 301-305: These are common knowledge things that should be removed.
Response 35: Many thanks for this comment. We have deleted them according to the Reviewer’s comments.
Point 36: Line 307: “This is because asphalt binder is a typical viscous-elastic material. It changes from elastic state into non-Newtonian fluid state as temperature increases.” This sentence should be removed. In this case, G* is not decreasing for this reason. Its reduction is due to the loss of the glass state of asphalt binder (see 10.1021/acs.energyfuels.7b03058 for basic knowledge).
Response 36: Many thanks for this comment. We have made correction according to the Reviewer’s comments, and referenced this article.
Point 37: Line 329: asphalt improves -> asphalt that improves.
Response 37: Many thanks for this comment. We have made a correction according to the Reviewer’s comments.
Point 38: Line 331: property -> properties.
Response 38: Many thanks for this comment. We have made a correction according to the Reviewer’s comments.
Point 39: Line 334: Please decode the abbreviation SHRP.
Response 39: Many thanks for this comment. We have made a corrections in the article. SHRP correct to strategic highway research program.
Point 40: Line 338: the average G* values -> the average G*/sinδ values?
Response 40: Thanks for the careful review, G* here is not edition error. It refers to the dynamic shear module in Figure 11, and it mentioned here to compare with G*/sinδ.
Point 41: Table 5. Title. Please decode the abbreviation PG.
Response 41: Many thanks for this comment. We have made a corrections in the article. PG correct to performance grade.
Point 42: Lines 391-393: “The changing of absorption peak intensity and emerge of new absorption peaks indicate chemical reactions between PCB-NMF, compatibilizer and matrix asphalt.” No, it does not. To prove this, the change in spectra during temperature treatment should be compared. This sentence should be removed as speculation.
Response 42: Many thanks for this comment. We have removed this sentence.
Point 43: Lines 395-398. This is speculation. It should be removed. Carbonyl groups cannot react with epoxy groups, which are also not present in the crosslinked resin.
Response 43: Many thanks for this comment. We have removed this sentence.
Point 44: Lines 399-412. This part should be removed because the DSC curves are not taken correctly. They should be obtained from low negative temperatures. Those that the authors show are not peaks caused by transitions but an instrumental error due to the calorimeter going into the measurement mode.
Response 44: Many thanks for this comment. We have removed this part.
Point 45: Lines 429-432. This should be removed as nothing more than pure speculation.
Response 45: Many thanks for this comment. We have removed it.
We tried our best to improve the manuscript and made some changes in the manuscript. These changes will not influence the content and framework of the paper. And here we did not list the changes but also can been seen under track changes in revised paper. In addition, we also have carefully checked through the whole manuscript and corrected some grammar mistakes, and asked a professor with UK Ph.D in our college and a native speaker to modify the English to improve the quality of the article..
We appreciate for Reviewers’ warm work earnestly, and hope that the correction will meet with approval.
Once again, thank you very much for your comments and suggestions.
Yours sincerely,
Yu Sun

Reviewer 2 Report
Thank you for the manuscript. I have a few suggestions to improve this manuscript.
1) Graphical abstract needs correction: RV measures high temperature properties not DSR. Points 1 and 2 are described confusingly.
2) Line 51 and 54 is too long to be understood.
3) Few terminologies in the literature review, taken from other researchers, can be confusing for the reader e.g PI, T800, and T1.2. Also RPG, CR, CRT. I suggest defining them.
4) The following paper on plastic can be useful as it provides information about plastic waste, especially about the thermal stability of plastic modified bitumen. https://bjrbe-journals.rtu.lv/article/view/3990
5) This is not clear "glass fiber is grounded into powder with no difference from glass powder.”
6) There are few typos e.g. line 115, line 120
7) The novelty of this research is not clear. Please improve it.
8) What is the source of technical requirements?
9) Figures 1 and 2 can be committed because they don’t provide any useful information. The materials are clear from figure 3.
10) What is the rationality of using 60% tung oil and 40% glycerin as compatibilizer?
11)nLine 169-173 needs to be improved and reference is required. Also, see the line 197-200.
12) "At the same time, penetration and SPD declined about 6.5% 168 and 17%, respectively.” Please check for the decline from 0 to 8%. Also, check line 174-175.
13) A certain amount of PCB-NMF, please provide the content.
14) 154- 158, aren’t these part of storage stability? The storage stability has been discussed later. It can be confusing for the reader. This is because the determination of optimum compatibilizer content is part of the results. It would be better to move this to the result section.
In section 2.3, the SEM discussion is ok. However, discussion of Figure 5 seems to be problematic. The rationale for 8% content is also not clear. I suggest revising it.
I do not understand the need for the heating rate in the DSR test.
I would suggest authors acknowledge the limitations of low temperature properties in conclusion 3.
Author Response
Response to Reviewer 1 Comments
Dear Reviewers:
Thank you for your letter and for the reviewers’ comments concerning our manuscript entitled “Recycling Non-Metallic Powder of Waste Printed Circuit Board in Improving the Performance of Asphalt Material”. Those comments are all valuable and very helpful for revising and improving our paper, as well as the important guiding significance to our researches. We have studied comments carefully and have made correction which we hope meet with approval. The main corrections in the paper and the responds to the reviewer’s comments are as following:
Responds to the reviewer’s comments:
Point 1: Graphical abstract needs correction: RV measures high temperature properties not DSR. Points 1 and 2 are described confusingly.
Response 1: Many thanks for this comment. It was decided to delete Graphic abstract because its representation was not clear enough and Figures 1 and 2 can also illustrates the entire flow of the study.
Point 2: Line 51 and 54 is too long to be understood.
Response 2: Many thanks for this comment. We have made an adjustment according to the Reviewer’s comments.
Point 3: Few terminologies in the literature review, taken from other researchers, can be confusing for the reader e.g PI, T800, and T1.2. Also RPG, CR, CRT. I suggest defining them.
Response 3: Many thanks for this comment. We have made correction according to the Reviewer’s comments.
Point 4: The following paper on plastic can be useful as it provides information about plastic waste, especially about the thermal stability of plastic modified bitumen. https://bjrbe-journals.rtu.lv/article/view/3990
Response 4: Thanks for recommendation. We have referenced this article in the introduction.
Point 5: This is not clear "glass fiber is grounded into powder with no difference from glass powder.”
Response 5: Many thanks for this comment. We have made an adjustment according to the Reviewer’s comments.
Point 6: There are few typos e.g. line 115, line 120
Response 6: Many thanks for this comment. We have made a correction according to the Reviewer’s comments.
Point 7: The novelty of this research is not clear. Please improve it.
Response 7: Thanks for your advice, we would make effort to do the research with more novelty next.
Point 8: What is the source of technical requirements?
Response 8: Many thanks for this comment. The source of technical requirements is from Chinese Standard Test Methods of Bitumen and Bituminous Mixtures for Highway Engineering. We have cited it in the article.
Point 9: Figures 1 and 2 can be committed because they don’t provide any useful information. The materials are clear from figure 3.
Response 9: Many thanks for this comment. We have removed these two pictures.
Point 10: What is the rationality of using 60% tung oil and 40% glycerin as compatibilizer?
Response 10: Many thanks for this comment. We have added the reason of why using 60% tung oil and 40% glycerin as compatibilizer.
Point 11: Line 169-173 needs to be improved and reference is required. Also, see the line 197-200.
Response 11: Many thanks for this comment. We have removed these two parts.
Point 12: "At the same time, penetration and SPD declined about 6.5% 168 and 17%, respectively.” Please check for the decline from 0 to 8%. Also, check line 174-175.
Response 12: Many thanks for this comment. The data was edited with errors and corrections have been made.
Point 13: Response to comment: A certain amount of PCB-NMF, please provide the content.
Response 13: Many thanks for this comment. The specific dosage of PCB-NMF here has been stated.
Point 14: 154- 158, aren’t these part of storage stability? The storage stability has been discussed later. It can be confusing for the reader. This is because the determination of optimum compatibilizer content is part of the results. It would be better to move this to the result section.
Response 14: Many thanks for this comment. We have made an adjustment.
Point 15: In section 2.3, the SEM discussion is ok. However, discussion of Figure 5 seems to be problematic. The rationale for 8% content is also not clear. I suggest revising it.
Response 15: Many thanks for this comment. Same as comment 12, the data was edited with errors and corrections have been made.
Point 16: I do not understand the need for the heating rate in the DSR test.
Response 16: Thanks for your comments. It was confirmed that this parameter is indeed meaningless here, and the heating rate has been deleted.
Point 17: I would suggest authors acknowledge the limitations of low temperature properties in conclusion 3.
Response 17: Many thanks for this comment. We have adjustment the statement in conclusion.
We tried our best to improve the manuscript and made some changes in the manuscript. These changes will not influence the content and framework of the paper. And here we did not list the changes but also can been seen under track changes in revised paper. In addition, we also have carefully checked through the whole manuscript and corrected some grammar mistakes, and asked a professor with UK Ph.D in our college and a native speaker to modify the English to improve the quality of the article..
We appreciate for Reviewers’ warm work earnestly, and hope that the correction will meet with approval.
Once again, thank you very much for your comments and suggestions.
Yours sincerely,
Yu Sun

Round 2
Reviewer 1 Report
The authors were very careful and correct in making changes within the manuscript. In my opinion, it can be published in Materials.
Author Response
Dear Reviewers:
Thank you for your patient review again. By studying the comments and making correction, the logic and the presentation of our article is much clearer. Thank you also for your affirmation of the article's revision and the article you recommended.
Once again, thank you very much for your comments and suggestions.
Yours sincerely,
Yu Sun
Reviewer 2 Report
Thank you for the significantly improved manuscript. There are few additional suggestions from readability point of view.
In Figure 3c, the ductility properties have enhanced even if the value is not maximum, the authors can adjust the statement appropriately. Currently it is giving the impression that ductility has not improved.
Please check spelling of effect on line 243 and 297
Line 338: "Conventional properties” heading is also ok. Similarly, you may also refer use heading Rheological properties at line 385. At line 370 “Storage stability”
Line 242: Instead of "Electron microscope scanning test” please use “Scanning Electron Microscopy” similarly on line 273 use “Bending beam rheology"
Description of Figure 4 is too long, certain information can be omitted. Please check the magnification level on SEM image with the description of figure.
Author Response
Dear Reviewers:
Thank you for your patient review again. By studying the comments and making correction, the logic and the presentation of our article is much clearer. The main corrections in the paper and the responds to the reviewer’s comments are as following:
Responds to the reviewer’s comments:
Point 1: In Figure 3c, the ductility properties have enhanced even if the value is not maximum, the authors can adjust the statement appropriately. Currently it is giving the impression that ductility has not improved.
Response 1: Many thanks for this comment. We have adjusted the statement to make the difference of ductility more obverious.
Point 2: Please check spelling of effect on line 243 and 297
Response 2: Many thanks for this comment. We have carefully checked this part and corrected the word in 2.3. (1) and 2.3. (6).
Point 3: "Conventional properties” heading is also ok. Similarly, you may also refer use heading Rheological properties at line 385. At line 370 “Storage stability”
Response 3: Many thanks for this comment. Another reviewer had been confused with "Conventional properties” and thought that this description was not clear enough. So we made this adjustment, if it is ok, here we still use “Basic properties ananlysis” for heading.
Point 4: Instead of "Electron microscope scanning test” please use “Scanning Electron Microscopy” similarly on line 273 use “Bending beam rheology"
Response 4: Many thanks for this comment. We have made corrections, and we corrected beam bending rheology to bending beam rheology in whole article.
Point 5: Description of Figure 4 is too long, certain information can be omitted. Please check the magnification level on SEM image with the description of figure.
Response 5: Many thanks for this comment. We have simplified the description of the image, and made a correction of magnification level.
We appreciate for Reviewers’ warm work earnestly, and hope that the correction will meet with approval.
Once again, thank you very much for your comments and suggestions.
Yours sincerely,
Yu Sun